# Adaptive Weighted Error-Correction Method Based on the Error Distribution Characteristics of Multi-Channel Alignment

**DOI:** 10.3390/s24092756

**Published:** 2024-04-26

**Authors:** Peiyu Song, Weibo Wang, Biwei Wu, Limin Zou, Tianpeng Zhan, Jiubin Tan, Xuemei Ding

**Affiliations:** 1Center of Ultra-Precision Optoelectronic Instrument Engineering, Harbin Institute of Technology, Harbin 150001, China; 20b901008@stu.hit.edu.cn (P.S.); bewaywu@hit.edu.cn (B.W.); zoulimin@hit.edu.cn (L.Z.); 20s001060@stu.hit.edu.cn (T.Z.); jbtan@hit.edu.cn (J.T.); xmding@hit.edu.cn (X.D.); 2Key Lab of Ultra-Precision Intelligent Instrumentation, Harbin Institute of Technology, Ministry of Industry and Information Technology, Harbin 150001, China

**Keywords:** wafer alignment, mark asymmetry, multi-channel alignment sensor, error distribution characteristic, multi-channel weighted method

## Abstract

As process nodes of advanced integrated circuits continue to decrease below 10 nm, the requirement for overlay accuracy is becoming stricter. The alignment sensor measures the position of the alignment mark relative to the wafer; thus, sub-nanometer alignment position accuracy is vital. The Phase Grating Alignment (PGA) method is widely used due to its high precision and stability. However, the alignment error caused by the mark asymmetry is the key obstacle preventing PGA technology from achieving sub-nanometer alignment accuracy. This error can be corrected using many methods, such as process verification and multi-channel weighted methods based on multi-diffraction, multi-wavelength and multi-polarization state alignment sensors. However, the mark asymmetry is unpredictable, complex and difficult to obtain in advance. In this case, the fixed-weight method cannot effectively reduce the alignment error. Therefore, an adaptive weighted method based on the error distribution characteristic of a multi-channel is proposed. Firstly, the simulation result proves that the error distribution characteristic of the multi-alignment result has a strong correlation with the mark asymmetry. Secondly, a concrete method of constructing weight values based on error distribution is described. We assume that the relationship between the weight value of each channel and the deviations of all channels’ results is second-order linear. Finally, without other prior process correction in the simulation experiment, the residual error’s Root Mean Square (RMS) of fixed weighted method is 14.0 nm, while the RMS of the adaptive weighted method is 0.01 nm, when dealing with five typical types of mark asymmetry. The adaptive weighted method exhibits a more stable error correction effect under unpredictable and complicated mark asymmetry.

## 1. Introduction

The integrated circuit (IC) is a key component of the information industry. Modern ICs are stacked by dozens of individual layers, and the accurate positioning of each layer, referred to as overlay accuracy, is essential for optimal performance. The resolution, overlay, and throughput are the three critical indicators of photolithography equipment [1,2]. The overlay error beyond the limitation may cause gross deviation of the IC components and devices, and lead to short circuits, open circuits and other problems [3,4,5]. According to the International Technology Roadmap for Semiconductors (ITRS) and other institutes, for process nodes of 10 nm and 7 nm, the overlay accuracy is required to be below 3 nm and 2 nm [6,7,8,9,10].

On-product overlay control is guaranteed by controlling alignment and overlay metrology, typically through feedforward and feedback control loops in customer fab [11]. The continuous reduction of alignment error is the key to ensuring the overlay accuracy. Dozens of alignment marks are distributed on the entire wafer, and each alignment mark’s position is measured by an alignment sensor to obtain wafer distortion. According to the wafer distortion, the exposure process parameters are adjusted to accurately transfer the circuit patterns from mask to wafer [12,13]. The overlay accuracy is generally 1/3 that of the process nodes, and the alignment accuracy is about 1/5~1/3 of the overlay accuracy. Therefore, the alignment accuracy should be less than 1 nm and 0.67 nm to ensure overlay accuracy for process nodes of 10 nm and 7 nm.

Phase Grating Alignment (PGA) mark position measurement technology [14] has high measurement accuracy, a fast measurement speed and is less affected by the measurement condition. Phase diffraction gratings are widely used as alignment marks in wafer alignment systems [15]. The wafer stage drives the alignment mark to scan under the sensor. The sensor illuminates the grating and collects the diffraction light to form the intensity signal. By analyzing the position-intensity signal, the PGA sensor is able to obtain the position of the alignment mark.

The PGA technology requires that the structure of the grating must be symmetrical [9]. Unfortunately, with the growing complexity of the pattern stacks and post-litho processing, the alignment mark’s profile incurs deformation through the semiconductor manufacturing process, such as etching, high temperature, annealing, chemical mechanical polishing (CMP), deposition, oxidation, and coating. The symmetric deformation will lead to the reduction of diffraction efficiency and the alignment signal’s SNR, and the asymmetric deformation will directly result in an alignment-position error. Generally, the etching and deposition process cause side-wall asymmetry, the polishing process causes top asymmetry, and the coating and exposure process lead to the bottom asymmetry [16,17,18,19,20,21]. The asymmetry of the alignment marks cause errors in the alignment results and lead to overlay errors unrelated to wafer distortion [22]. In order to improve the overlay accuracy, the effect of alignment mark asymmetry must be removed [13].

The measurement error resulting from the asymmetric deformation of the grating profile is a very important research topic in the field of semiconductors. Researchers have proposed many methods to address the error caused by the asymmetry overlay mark. Jiahao Zhang proposed a high-precision X-ray-based overlay metrology using reciprocal space-slicing analysis (RSS). This method has demonstrated robustness against asymmetric overlay marks [23]. However, the wafer surface is coated with photoresist during the alignment process. X-ray-based metrology may accidentally expose the photoresist. The optimized overlay mark design method proposed by Hung-Chih Hsieh aims to minimize the overlay error caused by asymmetric overlay marks [24]. Their findings indicate that ensuring the linewidth of the bottom overlay mark matches the pitch of the top overlay mark effectively reduces the overlay error. Hung-Chih Hsieh proposed an innovative method for wavelength selection in overlay metrology that eliminates the need for wafers [25]. A calculation model was developed to account for the presence of asymmetric overlay marks, which revealed that additional diffraction intensities contribute to the overlay error. To quantify this effect, an asymmetry factor was introduced and simulation results were used to determine the optimized wavelength that minimizes the overlay error. Hung-Chih Hsieh proposed an innovative approach to enhance the accuracy of cross-layer misalignment (MA) measurement by introducing a multiple-wavelength MA measurement error model [26]. Furthermore, finite-difference time-domain (FDTD) simulations demonstrate the linear relationship between MA value and pattern center shift, validating its applicability. Additionally, a correction method based on this linear property is introduced. Yating Shi presents a multi-objective optimization approach to enhance the design of overlay targets in diffraction-based overlay (DBO) metrology, aiming to improve measurement precision, accuracy, and robustness against process variations and target deformation [27]. This approach relies on empirical DBO techniques that utilize specially designed targets for measuring overlay errors without the need to solve inverse problems, thereby enhancing real-time measurement capabilities.

Currently, several methods are employed to correct alignment position error, including the process verification method, multi-diffraction order-weighted method and multi-channel-weighted method. In the process verification method, the overlay error of the two layers is precisely measured using scanning electron microscopy (SEM) after lithography, and the alignment position error will be corrected in the subsequent process based on the measured overlay error [6,17,22]. However, since the mark asymmetries differ between wafers and exposure fields, even for the same batch and process, this method can not correct the alignment error stably. The process verification method is time-consuming, costly, destructive and lacks robustness [6,7]. This also indicates that it is not practical to obtain the asymmetry of the alignment mark directly before the alignment process.

Compared with the process verification method, the multiple alignment results weighted method can correct the alignment error in real-time. Guanghua Yang analyzed the effect of mark asymmetry on alignment position error using scalar diffraction theory [28], revealing that the alignment position error △*x* is a function of diffraction order *m*. For the same asymmetry, different order *m* has different △*x*. On this basis, the alignment position error can be reduced by the weighted sum of multiple alignment position results of different diffraction orders.

As a result of the increasing application of new materials in lithography, the alignment sensor has developed from single-wavelength to multi-wavelength, and then to multi-wavelength and multi-polarization state [29,30,31]. In 2017, ASML released the ORION alignment sensor on the NXT2000i lithography machine. Compared with the SMart Alignment Sensor Hybrid (SMASH) with four wavelengths and one polarization state, the ORION sensor utilizes four wavelengths and two polarization states to ensure the Signal-Noise-Ratio (SNR) is adequate when dealing with different lithography materials [32,33]. As a result, the alignment sensor can obtain eight alignment position results in different channels when scanning an alignment mark.

The alignment position results of each channel are not the same, since the alignment position errors are related to the measurement wavelengths and polarization states [6]. Juyou Du proposed a method that weighted the alignment position results of different channels to reduce the alignment position error [34,35]. However, this method only uses alignment position results in two channels, and the sum of weight values is not constrained to one, which artificially reduces the alignment position error. This method cannot be successfully applied to the actual alignment process. In 2018, ASML researchers presented the Optimal Color Weighting (OCW) method, which weights the alignment position results of the ORION sensor’s eight channels, with the sum of the eight weight values being one [22]. OCW is a multi-channel weighted method and it effectively reduced the alignment position error. In addition, the ASML researchers also proposed the Wafer Alignment Model Mapping (WAMM) method. WAMM is capable of reducing the contribution of mark asymmetry to an overlay by using a more optimal high-order wafer alignment recipe, and WAMM can be combined with OCW to further improve the overlay accuracy.

However, the weight values used by these methods will not be adjusted when the mark asymmetry changes. The OCW method conducts the experiment with a limited number of wafers. In the real FAB, the various mark asymmetries are unpredictable, complicated and are mixed together [11,16,22]. The weight values with the optimal correction effect change with the mark asymmetry; thus, these fixed weighted methods have weaknesses in process robustness and flexibility.

Therefore, we propose an adaptive weighted method that can adjust weight values according to the mark asymmetry. In the multi-channel alignment sensor, different channels’ alignment positions have different correlations and sensitivities to the mark asymmetry. The alignment position error distribution of multi-channel is strongly correlated with the mark asymmetry. In the proposed method, the weight values of each channel are calculated according to the error distribution characteristic. Therefore, when the mark asymmetry changes, the weight values of each channel can be adaptively adjusted to the optimal values in the proposed method.

## 2. Theoretical Model

Figure 1 presents the schematic diagram of a multi-channel alignment sensor. The alignment mark highlights periodic phase-diffraction grating. The laser with high spatial coherence is illuminated vertically at the alignment mark, and the positive and negative-diffraction-order light are collected and collimated by the objective. The self-referencing rotational shearing interferometer divides the diffraction light into two parts, with one part rotated by 180°, and the two parts combined afterward at the same optical axis. Hence, the +*m* and −*m* diffraction orders interfere with each other.

When the phase grating alignment mark on the wafer scans is relative to the alignment sensor, the phase φ of the positive and negative diffraction orders varies, and the intensity of the interference signal changes accordingly. In PGA technology, the positive and negative odd diffraction orders interfere with each other to generate a combination of multiple sinusoidal intensity signals, while the diffraction efficiency of zero and even diffraction order will only result in a decrease in the alignment measurement signal’s SNR [6,15,36].

In the present study, according to the grating equation, only ±1 diffraction orders can be collected by limiting the numerical aperture of the objective and the grating pitch. When the alignment mark is scanned relative to the sensor, the complex amplitude of +1 and −1 diffraction order can be calculated as:(1)$$E_{+1}=\left|E_{+1}\right|exp\left\{j\left[-\frac{2\pi }{P}bias+\varphi_{+1}\right]\right\}$$
(2)$$E_{-1}=\left|E_{-1}\right|exp\left\{j\left[\frac{2\pi }{P}bias+\varphi_{-1}\right]\right\}$$
where *E* is the complex amplitude, *P* is the pitch of the grating, and bias is the relative position of the alignment mark with respect to the alignment sensor’s optical axis [37]. Thus, the alignment measurement position-intensity signal can be calculated as
(3)$$I\left(\delta \right)={\left|E_{+1}\right|}^{2}+{\left|E_{-1}\right|}^{2}+2\left|E_{+1}\right|\left|E_{-1}\right|\cos\left[\frac{4\pi }{P}bias-\left(\varphi_{+1}-\varphi_{-1}\right)\right]$$

The position of the alignment mark is determined based on the phase of the Fourier transform of the intensity signal.

On a symmetric alignment mark, the alignment position results measured by all four wavelengths and two polarization states are the same, as the initial phase difference $\Delta \varphi =\varphi_{+1}-\varphi_{-1}$ is always zero. In contrast to the symmetric alignment mark, the initial phase difference of the asymmetric mark is non-zero and results in an alignment position error. The additional alignment signal is shifted in a phase which can no longer be distinguished from the actual physical shift of the mark when there is a mark asymmetry [6]. Moreover, for different wavelength and polarization states—that is, different *m*th measurement channels—the initial phase difference of the asymmetric mark is different, and can be expressed as $\Delta \varphi_{m}$. The alignment sensor in this study employs four wavelengths and two polarization states, enabling the acquisition of eight channels of alignment signals.

For convenience of expression, $x_{m}$ represents the alignment position calculated from the *m*th channel, $\Delta x_{m}$ is the alignment position error caused by mark asymmetry, and $x_{0}$ denotes the actual position of the alignment mark. This study focuses on the influence of mark asymmetry on the alignment measurement accuracy. The position measurement error caused by other factors is not considered. Therefore, the Equations (Equation 4) and (Equation 5) can be obtained as:(4)$$\Delta x_{m}=\frac{\Delta \varphi_{m}}{4\pi }P$$
(5)$$x_{m}=x_{0}+\Delta x_{m}$$

The results obtained from the COMSOL simulation, as detailed later in the paper, indicate that the alignment error varies across different channels for the same mark asymmetry. The weighted methods utilize the measurement information and sensitivity difference in multiple channels to obtain a fitting alignment position result, reducing the alignment error caused by mark asymmetry. The weight values of *m*th channel are denoted by $w_{m}$, and the error-corrected alignment position by weighted method is calculated as:(6)$$x=\sum_{m=1}^{8}w_{m}x_{m}$$

The substitution of Equation (Equation 5) into Equation (Equation 6) yields
(7)$$x=x_{0}\sum_{m=1}^{8}w_{m}+\sum_{m=1}^{8}w_{m}\Delta x_{m}$$

To ensure *x* is as close to $x_{0}$ as possible, the weight values should meet the following conditions:(8)$$\sum_{m=1}^{8}w_{m}=1$$
(9)$$\sum_{m=1}^{8}w_{m}\Delta x_{m}=0$$

Equation (Equation 8) represents a necessary condition, while Equation (Equation 9) aims to minimize the alignment measurement error. The above Equations illustrate the principle of alignment error correction by weighted method.

As Equation (Equation 8) represents a necessary condition, $w_{8}$ among the eight weight values in the equation can be expressed as
(10)$$w_{8}=1-\sum_{m=1}^{7}w_{m}$$

Then, Equation (Equation 9) can be expressed as
(11)$$\sum_{m=1}^{7}\left(\Delta x_{m}-\Delta x_{8}\right)w_{m}=-\Delta x_{8}$$

In the fixed weighted method, the optimal weight values $w_{m}$ are calculated based on the previous simulation and experimental data, and will not change during the alignment process. While the alignment mark asymmetry is unpredictable and complicated even in the same wafer, fixed weighted method cannot adjust the weight values to the optimal values in the real alignment process.

Given *N* groups of asymmetric alignment mark and corresponding multi-channel alignment position error data, the optimal weight value $w_{m}$ can be determined using the least squares method. The matrix representation of above process can be formulated as follows:(12)Aw=b

The size of matrix *A* is N×7, and the element corresponding to the position is $\Delta x_{m,r}-\Delta x_{8,r}$, where $m\in \left[1:7\right]$ and $r\in \left[1:N\right]$. *w* and *b* are 7×1 and N×1 column vectors composed of $w_{m}$ and $-\Delta x_{8}$.

In the proposed method, the weight values $w_{m}$ are designed to be calculated based on the alignment error distribution, that is, $\Delta x_{m}$ of each channel. Assuming that the weight value $w_{m}$ is a second-order function of the deviation $\Delta x_{m}^{\prime }$, the adaptive weighted method can be expressed as
(13)$$w_{m}=r_{m}+\sum_{i=1}^{8}\left[p_{m,i}{\left(\Delta x_{i}\right)}^{2}+q_{m,i}\Delta x_{i}\right]$$

Unfortunately, it is not possible to determine the true alignment mark position and the alignment error $\Delta x_{m}$ in the real alignment process in theory. Nevertheless, the deviation $\Delta x_{m}^{\prime }$ calculated by each channel’s result and the mean value of all channel’s result can also represent actual alignment position error distribution characteristics. Therefore, the deviation is used to calculate the weight values of each channel, which are expressed as
(14)$$\bar{x}=\frac{1}{8}\sum_{m=1}^{8}x_{m}$$
(15)$$\Delta x_{m}^{\prime }=x_{m}-\bar{x}$$
(16)$$w_{m}=r_{m}+\sum_{i=1}^{8}\left[p_{m,i}{\left(\Delta x_{i}^{\prime }\right)}^{2}+q_{m,i}\Delta x_{i}^{\prime }\right]$$

Similarly, by substituting Equation (Equation 16) into Equation (Equation 11), with *N* groups of asymmetric alignment marks and the corresponding multi-channel alignment position error data and least square method, the matrix expression can be obtained.
(17)Cy=b

The *C* matrix has *N* rows and M=7×8+7×8+7=119 columns, and *y* is a M×1 column vector composed of $r_{m}$, $p_{m,i}$ and $q_{m,i}$. The best estimates of the above parameters can be obtained.

## 3. Experiment Results and Analysis

In order to obtain better alignment accuracy and the positioning flexibility of the alignment mark, the current grating pitch of the alignment mark is getting smaller from 16 μm to 1 μm∼5 μm [16,38,39]. If the period of the grating is in the range of 2∼5 times the wavelength, the diffraction results calculated using scalar theory and vector theory have certain differences [40]. Although the calculation of scalar diffraction theory is faster, the grating pitch of the alignment mark is close to the illumination wavelength and the illumination beam has two polarization states, the precision of the scalar diffraction theory is not sufficient. The calculation method based on vector diffraction theory has better precision. Although the calculation based on vector diffraction theory takes more time, the data set required by the proposed method can be generated offline in advance, ensuring the efficiency of the alignment process.

We conducted COMSOL simulations to analyze the alignment position error across various channels subjected to different mark asymmetries. The COMSOL software (ver 6.0) is based on the finite element method for solving Maxwell equations and can be used to simulate the diffraction phenomena, such as diffraction efficiency and diffraction phase of the phase grating. To realistically mimic the lithography process, the simulated grating has a pitch of 2 μm, a phase depth of 160 nm, and a duty ratio of 0.5. Compared with 16 μm, the grating pitch of 2 μm reduces the effect of mark asymmetry, improves the alignment accuracy, and reduces the occupied area on the wafer [16,39]. The alignment mark material is set to Si, while the upper medium material is set to SiO_2_ with a thickness of 800 nm. The wavelengths are 532 nm, 633 nm, 775 nm and 850 nm, and the polarization states are set to TE and TM, which means the polarization direction is parallel and perpendicular to the grating vector. We consider the deformation model of one period as a unit and apply the periodic boundary condition (PBC) of Floquet period on both sides. To prevent the interference of scattered light, a perfect matching layer (PML) is employed at the bottom boundary. In order to achieve high accuracy, the subdivision size of the grid unit is set to 1/20 of the illumination wavelength. The “Parametric Sweep” function in the software can conveniently calculate the alignment errors under different asymmetry parameters, different incident wavelengths and different polarization states. The model of the asymmetric grating constructed in COMSOL and the diffraction electric field diagram are depicted in Figure 2.

In order to simplify the simulation modeling and calculation process, it is convenient to assume that the asymmetry of the alignment mark occurs only on one side of the profile when studying the influence of the mark asymmetry on the alignment accuracy. As a matter of fact, the mark asymmetry usually occurs on both sides of the grating profile, and the asymmetry is more complicated [41]. The structures of asymmetric alignment mark are presented in Figure 3. $P_{0}$ is the pitch of the grating, *d* is the phase depth, and *f* is the duty ratio. The types of mark asymmetry are defined as side wall, top, bottom, round corner, and wedge corner. The asymmetry parameters for these types are defined as *A*, *B*, *C*, *R* and *D*. According to previous research [35,41,42], it is reasonable to set the range of variation of these five parameters as 0∼80 nm.

When the optical axis of the alignment sensor coincides with the center of the symmetric alignment mark, the phase difference between positive and negative diffraction orders is zero. Therefore, the alignment error can be calculated according to the phase difference of the asymmetric mark’s positive and negative diffraction orders [6]. Figure 4 presents the simulated alignment error $\Delta x_{m}$ of *m*th channel. The multi-channel alignment error data of each asymmetric type form a data set, which provides a basis for the weighted method.

According to the fixed weighted method, a set of weight values can be calculated from each data set and the combination of all data sets, and the calculated optimal weight values $w_{m}$ are shown in Table 1. The weight values are applied to the corresponding data set, and the histogram distributions of residual error are depicted in Figure 5a–e.

The adaptive weighted method proposed in this paper uses a combination of five data sets, and one set of $p_{m,i}$ values is calculated. The combination of original alignment error data sets is corrected by the fixed weighted method with multiple sets of weight values coming from the individual data sets and the combined data sets. The combined error data sets are also corrected by the adaptive weighted method. The histogram distribution of residual alignment error is shown in Figure 6.

To evaluate the process robustness and flexibility, we use the root mean square (RMS) index of the residual alignment error to compare the correction effect of both methods. Table 2 shows the RMS of alignment error before and after being corrected by a fixed weighted and adaptive weighted method.

## 4. Discussion

It can be seen from the comparison of simulation results in Figure 4 that the alignment position error introduced by top asymmetry and bottom asymmetry is larger among the five mark asymmetries. The reason is that the top and bottom asymmetry mean the phase depths of the two sides of the grating have certain differences, and the phase information of the diffracted light is closely related to the phase depth. From Figure 4, it is evident that the alignment error resulting from bottom asymmetry demonstrates a non-linear and non-monotonic relationship with asymmetry parameters, which explains why the residue error of the bottom asymmetry in Figure 5 is larger than other asymmetry profiles.

The Figure 4 shows that the alignment error of multi-channel alignment is strongly correlated with the asymmetry of alignment marks. When the type and parameter of the mark asymmetry change, the numerical magnitude and plus and minus of the multi-channel measurement error deviation will change. It is feasible to infer the asymmetry of the alignment mark from the error distribution characteristics of the measurement results of multi-channel alignment sensor. The correlation between the two is the starting point of the proposed adaptive weighted method.

In the Figure 5, the histogram distribution of residuals error indicates that the fixed weighted method has a good error correction effect when the mark asymmetry is simple and appropriate weight values are used. However, it can be seen from Table 1 that the optimal weight values corresponding to different mark asymmetry types have significant differences. This indicates that when the mark asymmetry changes, the weight values of multi-channel should change accordingly.

In order to verify the robustness of the fixed weighted method, multi-channel alignment errors of all the mark asymmetry types are corrected by the optimal weight values corresponding to individual data sets and the combination of these data sets. The results are shown in Figure 6a–f. Similarly, multi-channel alignment errors of all the mark asymmetry types are corrected by the adaptive weighted, as shown in Figure 6g. In the Table 2, quantified RMS indicator shows the alignment error correction capabilities of each method.

The RMS of the original alignment data error is 49.8 nm. After being corrected by the optimal weight values corresponding to the single-asymmetry data set, the residual error decreases in some cases and increases in other cases. This indicates that the improper weight values will lead to an increase in the alignment error instead of a correction. The optimal weight values corresponding to the combination of these data sets reduce the RMS to 14.0 nm. This indicates that the effectiveness of the fixed weighted method can be improved by fitting more alignment error data related to different mark asymmetry.

However, on the other hand, the proposed adaptive weighted method results in the smallest residual errors of 0.01 nm with the same combined data set. This shows that the error correction effect of the adaptive weighted method is better than that of the fixed weighted method when the same alignment error data set is used in relation to different mark asymmetry. The proposed adaptive weighted method has better process robustness and flexibility when dealing with unpredictable and complicated mark asymmetry.

The coefficient $p_{m,i}$ in real process control can be determined by utilizing either more comprehensive simulation data or actual measurement data, which incorporate alignment and overlay information obtained from the scanner.

It is important to note that controlling the alignment accuracy requires a combination of various other methods and prior information, such as an alignment Sampling Scheme and layout Optimization (SSO) and advanced wafer alignment models based on wafer alignment model mapping (e.g., a higher-order wafer alignment model) [11,13]. On the other hand, the actual alignment accuracy can also be affected by other factors, such as internal noise of the sensor. The purpose of the above analysis is to show that the adaptive weighted method has a better error correction effect in comparison.

## 5. Conclusions

This paper proposes an adaptive weighted error correction method based on the error distribution characteristics of multi-channel alignment. The weighted sum of the alignment position from every channel can correct the alignment position error caused by the mark asymmetry to a certain extent. It can be seen that when the mark asymmetry changes, the optimal weight values will also change. Using the improper weight values will lead to an increase in the alignment error instead of a correction. On the other hand, the numerical simulation result shows that the alignment error of multi-channel alignment is strongly correlated with the asymmetry of alignment marks. In the proposed method, the weight values are not fixed, but are calculated based on the deviation of alignment errors. The error deviation represent the error distribution characteristic which has a strong relationship with the mark asymmetry. The proposed method allows the weight values to be automatically adjusted as the mark asymmetry changes. Further, the RMS indexes of each method are compared, and the result shows that the adaptive weighted method can provide a better error correction effect when dealing with unpredictable and complicated mark asymmetries with variable parameter values. The process robustness and flexibility of the proposed method are proved. Besides, the proposed method provides a more precise characterization of each wafer’s distortion, thereby ensuring wafer-to-wafer consistency of wafer alignment.

## Figures and Tables

**Figure 1 sensors-24-02756-f001:**
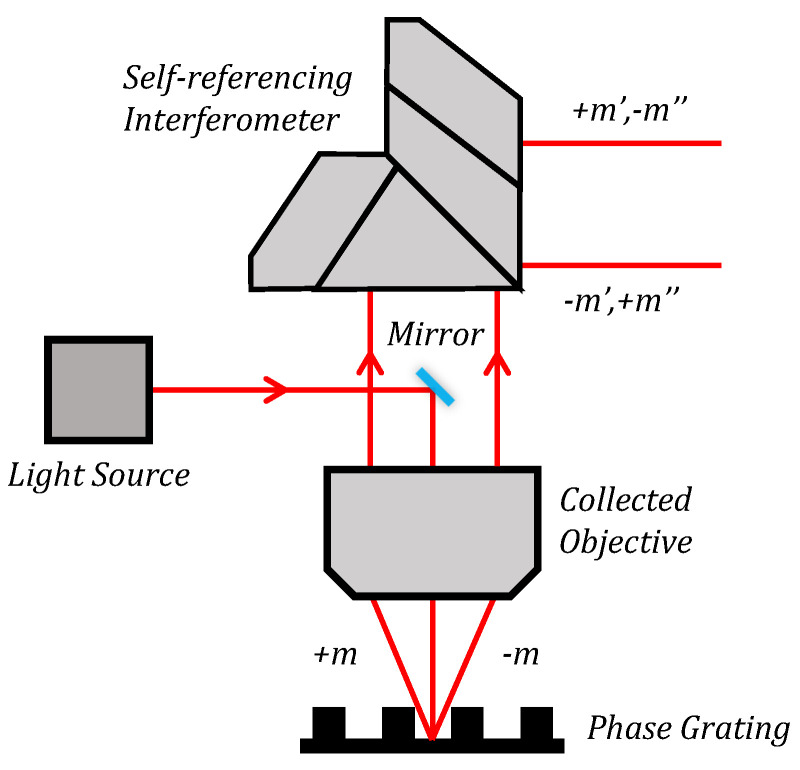
Schematic diagram of a multi-channel alignment sensor.

**Figure 2 sensors-24-02756-f002:**
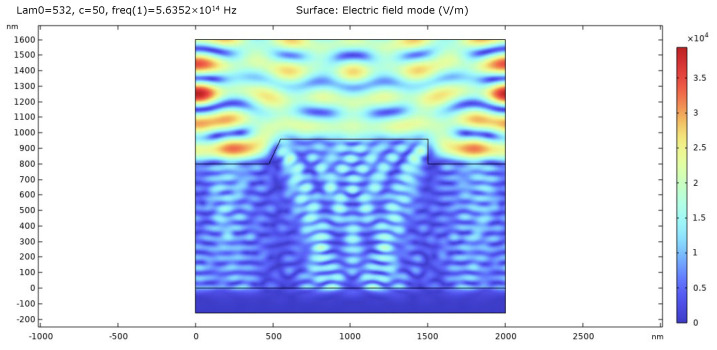
The diffraction electric field diagram of the asymmetric grating.

**Figure 3 sensors-24-02756-f003:**
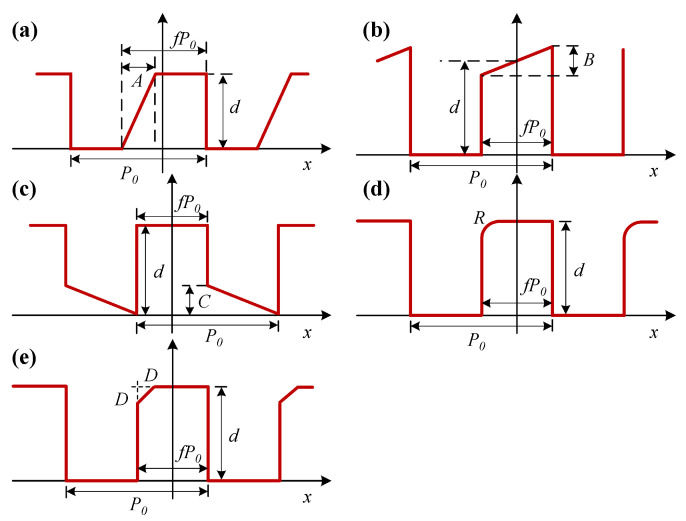
The structure of alignment mark with side wall (**a**), top (**b**), bottom (**c**), round corner (**d**), and wedge corner (**e**) asymmetry.

**Figure 4 sensors-24-02756-f004:**
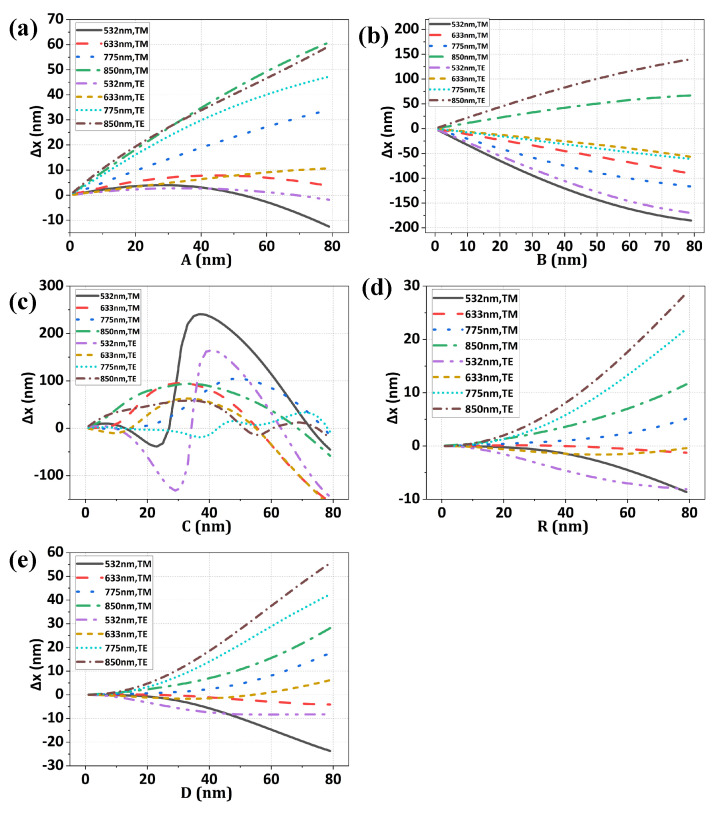
The alignment measurement error $\Delta x_{m}$ of *m*th channel is presented for side wall (**a**), top (**b**), bottom (**c**), round corner (**d**), and wedge corner (**e**) asymmetry and various parameters.

**Figure 5 sensors-24-02756-f005:**
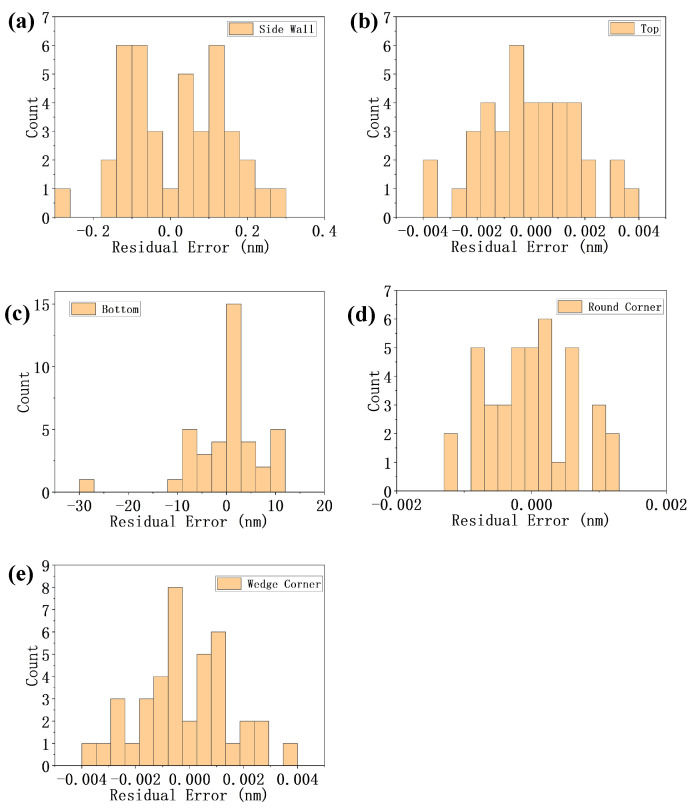
The residual error’s histogram of fixed weighted method applying the weight values of side wall (**a**), top (**b**), bottom (**c**), round corner (**d**), and wedge corner (**e**) to the corresponding data set.

**Figure 6 sensors-24-02756-f006:**
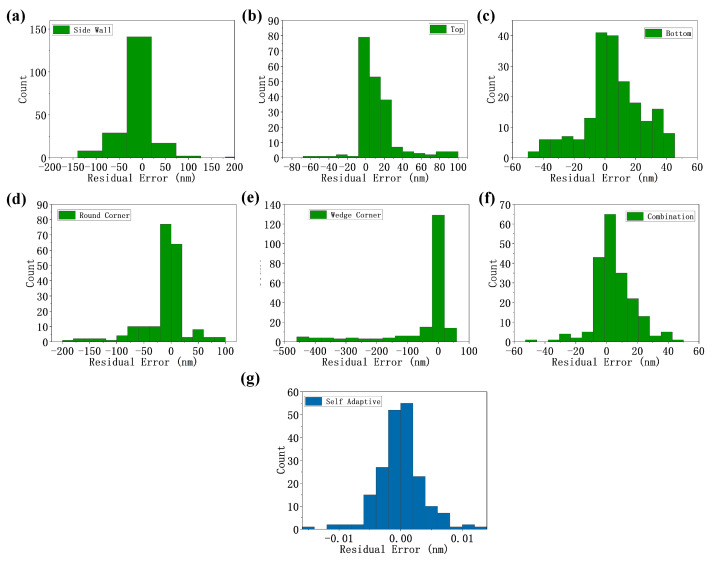
The residual error’s histogram obtained by correcting the error of the combined data using fixed weighted method based on side wall data (**a**), top data (**b**), bottom data (**c**), round corner data (**d**), wedge corner (**e**), combined data (**f**) and adaptive weighted method (**g**).

**Table 1 sensors-24-02756-t001:** The optimal weight values of the fixed weighted method calculated using different mark asymmetry.

Optimal Weight Values	Side Wall	Top	Bottom	Round Corner	Wedge Corner	Combination
$w_{1}$	0.82	0.10	0.04	−0.17	−0.29	0.08
$w_{2}$	−0.98	0.57	0.06	1.19	1.16	0.09
$w_{3}$	0.56	0.16	−0.14	−0.08	1.05	0.42
$w_{4}$	−1.36	−0.12	−0.10	−0.15	−0.58	−1.17
$w_{5}$	−1.25	0.04	0.01	0.09	−0.17	−0.04
$w_{6}$	2.23	−0.29	0.09	0.01	−0.55	0.47
$w_{7}$	0.55	−0.05	0.91	0.05	1.76	0.02
$w_{8}$	0.43	0.60	0.14	0.06	−1.38	1.13

**Table 2 sensors-24-02756-t002:** The RMS of alignment error before and after being corrected by fixed weighted and adaptive weighted methods.

	Original	Side Wall	Top	Bottom	Wedge Corner	Round Corner	Combination	Self-Adaptive
RMS (nm)	49.8	44.9	25.5	19.9	128.4	42.6	14.0	0.01

## Data Availability

Data underlying the results presented in this paper are not publicly available at this time, but may be obtained from the authors upon reasonable request.

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
