# Peer review of "Adaptive Weighted Error-Correction Method Based on the Error Distribution Characteristics of Multi-Channel Alignment"

_sensors, 2024, doi:10.3390/s24092756_

Round 1

Reviewer 1 Report

Comments and Suggestions for Authors

The manuscript " Process-robust alignment error correction method based on self-adaptive weighted fusion" by PEIYU SONG et al. describes a self-adaptive method for correcting wafer alignment errors. In the proposed approach, the weight is not a fixed value but is calculated based on the error distribution characteristics. Numerical simulations and results were performed.

The content of this manuscript fits in the scope of Sensors. However, after reviewing the content, I have some questions that I would like to ask the authors to address in a revised version:

1.     In the introduction paragraph, the authors mentioned that “the overlay accuracy is required below 3nm and 2nm”. So, what is the “alignment accuracy” under these overlay accuracy requirements?

2. The references provided by the author in the article have not been updated. References related to overlay and alignment from 2022 to 2024 should be added. Such as the research by Hung-Chih Hsieh, JIAHAO Zhang and XIUGUO Chen et al. In particular, Hung-Chih Hsieh's article on Optics and Lasers in Engineering provides the author with a reference for the simulation of overlay target profile asymmetry. 

3.     The sentence above the equation (6), “Substituting Equation 6 into Equation 5”. It should be “Substituting Equation 5 into Equation 6”.

4.     The authors should explain more clearly the advantage of equation (12), since it just like using all the alignment positions xm to calculate the weight values. It seems reasonable, but I wonder whether there is another calculation model for the weight values, which could give better results than this method and then be another paper.

5.     This paper only gave the numerical simulation. There are no lithography experiment results to verify the method. How to apply this method to the production line or real process control. Since we did not know the real asymmetry profile.

6.     Figures 3, 4 and 5 are too small and hard to read.

7. The authors should explain why the residue error of the bottom asymmetry in Figure 4 is so large than other asymmetry profiles.

8. In the process robustness simulation, the authors should consider the film thickness variation.

9. The most important for the scanner alignment is wafer-to-wafer consistency, how the authors proof that this method is suitable for this concern.

Comments on the Quality of English Language

Non.

Author Response

  1. In the introduction paragraph, the authors mentioned that “the overlay accuracy is required below 3nm and 2nm”. So, what is the “alignment accuracy” under these overlay accuracy requirements?

According to the proportional relationship between process nodes, overlay accuracy and alignment accuracy, the alignment accuracy should be less than 1 nm and 0.67 nm to ensure overlay accuracy for process nodes of 10 nm and 7 nm.

  1. The references provided by the author in the article have not been updated. References related to overlay and alignment from 2022 to 2024 should be added. Such as the research by Hung-Chih Hsieh, JIAHAO Zhang and XIUGUO Chen et al. In particular, Hung-Chih Hsieh's article on Optics and Lasers in Engineering provides the author with a reference for the simulation of overlay target profile asymmetry.

Thank you for your valuable suggestions regarding the references. These studies will greatly enhance the comprehensiveness of this paper's research. I have meticulously summarized and analyzed the pertinent references, which have been seamlessly incorporated into the manuscript.

Here is the discuss about these references in the manuscript.

“The measurement error resulting from the asymmetric deformation of the grating profile are very important research topic in the field of semiconductor. Researchers have proposed many methods to address the error caused by the asymmetry overlay mark. Jiahao Zhang proposed a high-precision x-ray-based overlay metrology using reciprocal space slicing analysis (RSS). This method has demonstrated robustness against asymmetric overlay marks. However, the wafer surface is coated with photoresist during the alignment process. X-ray based metrology may accidentally expose the photoresist. The optimized overlay mark design method proposed by Hung-Chih Hsieh aims to minimize the overlay error caused by asymmetric overlay marks. Their findings indicate that ensuring the linewidth of the bottom overlay mark to match the pitch of the top overlay mark effectively reduces the overlay error. Hung-Chih Hsieh proposed an innovative method for wavelength selection in overlay metrology that eliminates the need for wafers. A calculation model was developed to account for the presence of asymmetric overlay marks, which revealed that additional diffraction intensities contribute to the overlay error. To quantify this effect, an asymmetry factor was introduced and simulation results were used to determine the optimized wavelength that minimizes the overlay error. Hung-Chih Hsieh proposed an innovative approach to enhance the accuracy of cross-layer misalignment (MA) measurement by introducing a multiple-wavelength MA measurement error model. Furthermore, finite-difference time-domain (FDTD) simulations demonstrate the linear relationship between MA value and pattern center shift, validating its applicability. Additionally, a correction method based on this linear property is introduced. Yating Shi presents a multi-objective optimization approach to enhance the design of overlay targets in diffraction-based overlay (DBO) metrology, aiming to improve measurement precision, accuracy, and robustness against process variations and target deformation. This approach relies on empirical DBO techniques that utilize specially designed targets for measuring overlay errors without the need to solve inverse problems, thereby enhancing real-time measurement capabilities.”

  1. The sentence above the equation (6), “Substituting Equation 6 into Equation 5”. It should be “Substituting Equation 5 into Equation 6”.

This issue has been revised in the manuscript.

  1. The authors should explain more clearly the advantage of equation (12), since it just like using all the alignment positions xm to calculate the weight values. It seems reasonable, but I wonder whether there is another calculation model for the weight values, which could give better results than this method and then be another paper.

The equation (12) algorithm is based on the concept of a fully connected neural network, where each neuron in the subsequent layer is linked to every neuron in the preceding layer through weighted connections. Following training with datasets, the weight values within the network are established.

Recently, I attempted to model the relationship between weight values and alignment measurement deviations of each channel using a second-order polynomial, which can enhance the effectiveness of error correction. This section has been updated in the manuscript to reflect these improvements.

  1. This paper only gave the numerical simulation. There are no lithography experiment results to verify the method. How to apply this method to the production line or real process control. Since we did not know the real asymmetry profile.

The adaptive weighted method proposed in this paper utilizes the COMSOL software to simulate the alignment measurement error under various types of asymmetric gratings. Subsequently, the coefficient pm,i of the proposed method is computed based on these data, enabling correction of the alignment error.

In the real process control, the coefficient pm,i can be calculated using either more detailed simulation data or real measurement data that includes alignment and overlay data from the scanner.

The proposed method enables adaptive calculation of optimal weight values for multi-channel alignment sensors by utilizing the differences between multi-channel alignment measurement results without requiring the knowledge of the real asymmetric grating profile. Consequently, a higher level of accuracy is achieved through a weighted sum of each channel's alignment results.

  1. Figures 3, 4 and 5 are too small and hard to read.

The font size of these images has been adjusted.

  1. The authors should explain why the residue error of the bottom asymmetry in Figure 4 is so large than other asymmetry profiles.

From the FIG. 3, it can be seen that the alignment error resulting from bottom asymmetry exhibits a non-linear and non-monotonic relationship with asymmetry parameters, which poses challenges in error correction compared to other forms of asymmetry.

  1. In the process robustness simulation, the authors should consider the film thickness variation.

In the alignment measurement, the asymmetry between positive and negative diffraction light (phase and intensity) constitutes the alignment measurement signal. A uniform change in film thickness does not induce any asymmetry of diffraction light or lead to alignment measurement errors.

  1. The most important for the scanner alignment is wafer-to-wafer consistency, how the authors proof that this method is suitable for this concern.

In order to maintain the wafer-to-wafer consistency of wafer alignment, it is essential to accurately measure the wafer distortion. Due to the complexity of the lithography process, the asymmetric profile of the alignment mark varies among different wafers, resulting in uncertain alignment error. The method proposed in this paper can effectively correct alignment errors arising from mark asymmetry and provide a more precise description of each wafer's distortion, thereby ensuring wafer-to-wafer consistency of wafer alignment.

Reviewer 2 Report

Comments and Suggestions for Authors

The manuscript presents a well-structured and technically sound approach to addressing alignment errors in semiconductor manufacturing processes. The introduction effectively sets the context by highlighting the importance of overlay accuracy and the challenges posed by mark asymmetry. The theoretical model section provides a detailed explanation of the proposed adaptive weighted error correction method, supported by relevant equations and theoretical frameworks. The experimental results and analysis section offers comprehensive insights into the performance of the proposed method, with clear interpretation of simulation data and comparison with existing approaches. The discussion section effectively discusses the implications of the findings, emphasizing the superiority of the adaptive approach. Lastly, the conclusions succinctly summarize the key findings and highlight the practical implications of the proposed method. Overall, the manuscript demonstrates a thorough understanding of the problem domain and presents a novel and promising solution with clear potential for application in semiconductor manufacturing. However, mandatory revisions and clarifications should be done to further strengthen the manuscript.

1. The usage of abbreviations should be standardized, for example:

---The “RMS” is used in the Abstract but not defined.

---In the text, the “SNR” is used in Line 54 but defined in Line 83.

2. The quality of figures should be improved. In Figures 3 and 5, please improve the font size.

3. Could the authors further specify the criteria used to define the range of asymmetry parameters A, B, C, R, and D for the different types of mark asymmetry?

4. Could the authors provide more detailed descriptions of the COMSOL simulation setup to facilitate reproducibility and understanding?

5. Please further check the manuscript to avoid typos such as the missing spaces in “16μm” and “15μm” in Line 184.

Comments on the Quality of English Language

Moderate editing of English language required

Author Response

The manuscript presents a well-structured and technically sound approach to addressing alignment errors in semiconductor manufacturing processes. The introduction effectively sets the context by highlighting the importance of overlay accuracy and the challenges posed by mark asymmetry. The theoretical model section provides a detailed explanation of the proposed adaptive weighted error correction method, supported by relevant equations and theoretical frameworks. The experimental results and analysis section offers comprehensive insights into the performance of the proposed method, with clear interpretation of simulation data and comparison with existing approaches. The discussion section effectively discusses the implications of the findings, emphasizing the superiority of the adaptive approach. Lastly, the conclusions succinctly summarize the key findings and highlight the practical implications of the proposed method. Overall, the manuscript demonstrates a thorough understanding of the problem domain and presents a novel and promising solution with clear potential for application in semiconductor manufacturing. However, mandatory revisions and clarifications should be done to further strengthen the manuscript.

  1. The usage of abbreviations should be standardized, for example:

---The “RMS” is used in the Abstract but not defined.

---In the text, the “SNR” is used in Line 54 but defined in Line 83.

This issue has been revised in the manuscript

  1. The quality of figures should be improved. In Figures 3 and 5, please improve the font size.

The font size of these images has been adjusted.

  1. Could the authors further specify the criteria used to define the range of asymmetry parameters A, B, C, R, and D for the different types of mark asymmetry?

The following references provide support for the parameter selection range of this paper, although they are not direct arguments and evidences about the degree of marking asymmetry and parameter range.

“Du, J.; Dai, F.; Bu, Y.; Wang, X. Calibration method of overlay measurement error caused by asymmetric mark. Applied Optics 2018, 57, 9814–9821.”

“Pistor, T.V.; Socha, R.J. Rigorous electromagnetic simulation of stepper alignment. In Proceedings of the SPIE Advanced Lithography, 2002.”

“Li, L.; Chen, C.; Zeng, H.; Hu, S.; Zhang, L.; Su, Y.; Wei, Y.; Ye, T.; Wang, Y.; Xue, J. Analysis of diffraction-based wafer alignment rejection for thick aluminum process. Journal of Vacuum Science And Technology B 2022, 40, –.”

  1. Could the authors provide more detailed descriptions of the COMSOL simulation setup to facilitate reproducibility and understanding?

The “Parametric Sweep” function in the software can conveniently calculate the alignment errors under different asymmetry parameters, different incident wavelengths and different polarization states.

The model of the asymmetric grating built in COMSOL and the diffraction electric field diagram of the asymmetric grating are shown in the manuscript.

  1. Please further check the manuscript to avoid typos such as the missing spaces in “16μm” and “1∼5μm” in Line 184.

This issue has been revised in the manuscript.

Round 2

Reviewer 1 Report

Comments and Suggestions for Authors

All of my questions were well answered by the authors.

No more questions to be asked, and I think this manuscript could be accepted in its current form.

Comments on the Quality of English Language

No.

Author Response

Thank you very much for your comments and suggestions on my manuscript.

Reviewer 2 Report

Comments and Suggestions for Authors

My previous concern has been solved. However, the newly added Fig. 2 has obvious Chinese characters on the top. I'm sorry but this revised version is not inadequacy for publication. The authors should be more serious about the manuscript.

Author Response

I feel very sorry for this mistake, which should not have happened in this manuscript. I have revised it and examined the manuscript more carefully. I will pay more attention to the inspection and verification in the future academic papers.